# Rev7 and 53BP1/Crb2 prevent RecQ helicase-dependent hyper-resection of DNA double-strand breaks

Bryan A Leland, Angela C Chen, Amy Y Zhao, Robert C Wharton, Megan C King*

Department of Cell Biology, Yale School of Medicine, New Haven, United States

**Abstract** Poly(ADP ribose) polymerase inhibitors (PARPi) target cancer cells deficient in homology-directed repair of DNA double-strand breaks (DSBs). In preclinical models, PARPi resistance is tied to altered nucleolytic processing (resection) at the 5' ends of a DSB. For example, loss of either 53BP1 or Rev7/MAD2L2/FANCV derepresses resection to drive PARPi resistance, although the mechanisms are poorly understood. Long-range resection can be catalyzed by two machineries: the exonuclease Exo1, or the combination of a RecQ helicase and Dna2. Here, we develop a single-cell microscopy assay that allows the distinct phases and machineries of resection to be interrogated simultaneously in living *S. pombe* cells. Using this assay, we find that the 53BP1 orthologue and Rev7 specifically repress long-range resection through the RecQ helicase-dependent pathway, thereby preventing hyper-resection. These results suggest that 'rewiring' of BRCA1-deficient cells to employ an Exo1-independent hyper-resection pathway is a driver of PARPi resistance.

DOI: https://doi.org/10.7554/eLife.33402.001

## Introduction

DNA repair is an essential process conserved throughout evolution and commonly disrupted in tumor cells (*Jeggo et al., 2016*). Many cancer treatments, including poly(ADP ribose) polymerase (PARP) inhibitors (PARPi), target DNA repair pathways to kill rapidly dividing, repair-deficient cells (*Farmer et al., 2005*; *Fojo and Bates, 2013*; *Lord et al., 2015*; *Mateo et al., 2015*). 5' end resection, which generates tracts of single-strand DNA (ssDNA) at DNA double-strand break (DSB) ends dictates repair pathway choice: blocking resection promotes canonical non-homologous end joining (typically in G1), while initiating resection commits a DSB to repair by homologous recombination (HR), usually in S/G2 (*Chapman et al., 2012*; *Hustedt and Durocher, 2016*; *Symington, 2016*). The resection machinery is tightly controlled at both the step of resection initiation (involving Ctp1/Sae2/CtIP and the MRN/X complex) and during long-range resection, which is mediated by two parallel pathways catalyzed by either the exonuclease Exo1 or the combination of a RecQ helicase and Dna2 (*Chen et al., 2012*; *Croteau et al., 2014*; *Nimonkar et al., 2011*; *Tkáč et al., 2016*; *Zimmermann et al., 2013*). Additional accessory factors play key roles as modulators of resection; for example, loss of Rev7/MAD2L2/FANCV, a small, multifunctional HORMA domain protein (*Bluteau et al., 2016*; *Rosenberg and Corbett, 2015*) derepresses resection inhibition (*Boersma et al., 2015*; *Xu et al., 2015*), thereby allowing HR-deficient, $Brca1^{-/-}p53^{-/-}$ cells to become resistant to PARPi. In human cells, Rev7 appears to act in concert with another inhibitor of resection, 53BP1(*Chapman et al., 2013*; *Ochs et al., 2016*; *Zimmermann et al., 2013*), loss of which is also sufficient to drive PARPi resistance (*Boersma et al., 2015*; *Bouwman et al., 2010*; *Jaspers et al., 2013*; *Xu et al., 2015*). Importantly, the mechanisms by which Rev7 and 53BP1 inhibit DSB end resection remain poorly understood. To gain insights into how resection is controlled, we have developed a single-cell microscopy-based assay capable of quantitatively measuring DSB end

*For correspondence:
megan.king@yale.edu

**eLife digest** Healthy cells can typically repair damage to their DNA with high accuracy, keeping their genetic code intact. In contrast, cancer cells often lose this ability. Inaccurate repair leads to more frequent DNA mutations, which can make a tumor more aggressive. However, DNA repair-deficient tumors can be targeted with cancer therapies, such as PARP inhibitors, which kill cells that do not have working DNA repair mechanisms. PARP inhibitors show great promise clinically, but unfortunately some tumor cells can become resistant to these treatments over time. Recent work has shown that resistance to PARP inhibitors is often caused by further alternations to DNA repair machineries.

Being able to visualize DNA repair in living cells is crucial to understanding this process and to find ways to improve cancer treatments. Previous studies have used repetitive DNA sequences called Lac operators (LacO) to visualize the dynamic behavior of DNA in live cells. Leland et al. have now adapted this system to watch individual DNA repair events in living yeast cells under the microscope. Their experiments reveal that when cells lose a single protein called Rev7, an early phase of DNA repair becomes hyperactive. Leland et al. traced the cause of this hyperactivity to an enzyme in the RecQ helicase family.

A RecQ helicase becoming hyperactive in cells lacking Rev7 could explain how some cancer cells become resistant to PARP inhibitor treatments. This information could help fine-tune future approaches to treating cancer. For example, using an inhibitor of RecQ helicase alongside PARP inhibitors may help block this type of resistance from developing in the first place. As well as potentially paving the way for better cancer treatments, this method of visualization could improve scientists' understanding of the basic processes of DNA repair.
DOI: https://doi.org/10.7554/eLife.33402.002

resection rates in the facile genetic model, *S. pombe*. Leveraging this assay, we find that Rev7 and the 53BP1 orthologue, Crb2, specifically inhibit the RecQ-helicase-dependent long-range resection pathway. Moreover, through derepression of RecQ helicases, *rev7Δ* or *crb2Δ* cells can achieve very fast resection rates (>20 kb/hr) – approximately twice as fast as Exo1-dependent long-range resection. As BRCA1 activity has been tied to Exo1-dependent long-range resection (*Tomimatsu et al., 2012*), our findings suggest that PARPi resistance can be driven by compensation through derepression of the RecQ-helicase-dependent resection pathway.

## Results

### A microscopy-based assay to measure the rate of long-range resection in single cells

In order to quantitatively measure initial steps in DSB processing in single, living cells, we developed a microscopy-based DSB end resection assay (*Figure 1A*). In this system, an ectopic 10.3 kb, 256-copy LacO array and adjacent HO endonuclease cut site (HOcs) are engineered at a euchromatic (but intergenic) region near *Mmf1* (*Figure 1—figure supplement 1*). A single, site-specific DSB is generated by regulating the expression of the HO endonuclease under the control of the Ura-inducible $P_{urg1lox}$ RMCE system (*Watson et al., 2008*; *2011*). The timing of on-target DSB events is visualized by the appearance of a Rad52(Rad22)-mCherry focus that co-localizes with Mmf1:LacO/LacI-GFP (*Figure 1B*). By tracking cell lineages, we see that HO endonuclease induction produces on-target Rad52-mCherry foci in S/G2 (G1 is very short in *S. pombe*) when the repair machinery is primed for HR (*Symington and Gautier, 2011*) (*Figure 1B*). Importantly, we do not observe loading of Rad52-mCherry at the LacO/LacI-GFP array in the absence of HO endonuclease expression (on-target Rad52 foci in <0.2% of uninduced cells, n = 657), suggesting that the LacO array is not sufficient to create a 'fragile site' in *S. pombe* (*Jacome and Fernandez-Capetillo, 2011*; *Saad et al., 2014*).

As resection proceeds, the LacO repeats become single-stranded, disrupting LacI-GFP binding and causing the intensity of the GFP focus to progressively decrease (*Bell and Lewis, 2001*) (*Figure 1B*, *Figure 1—figure supplement 2*). To verify that loss of GFP focus intensity reflects DSB end resection, we analyzed cells lacking Exo1, which catalyzes the majority of long-range resection

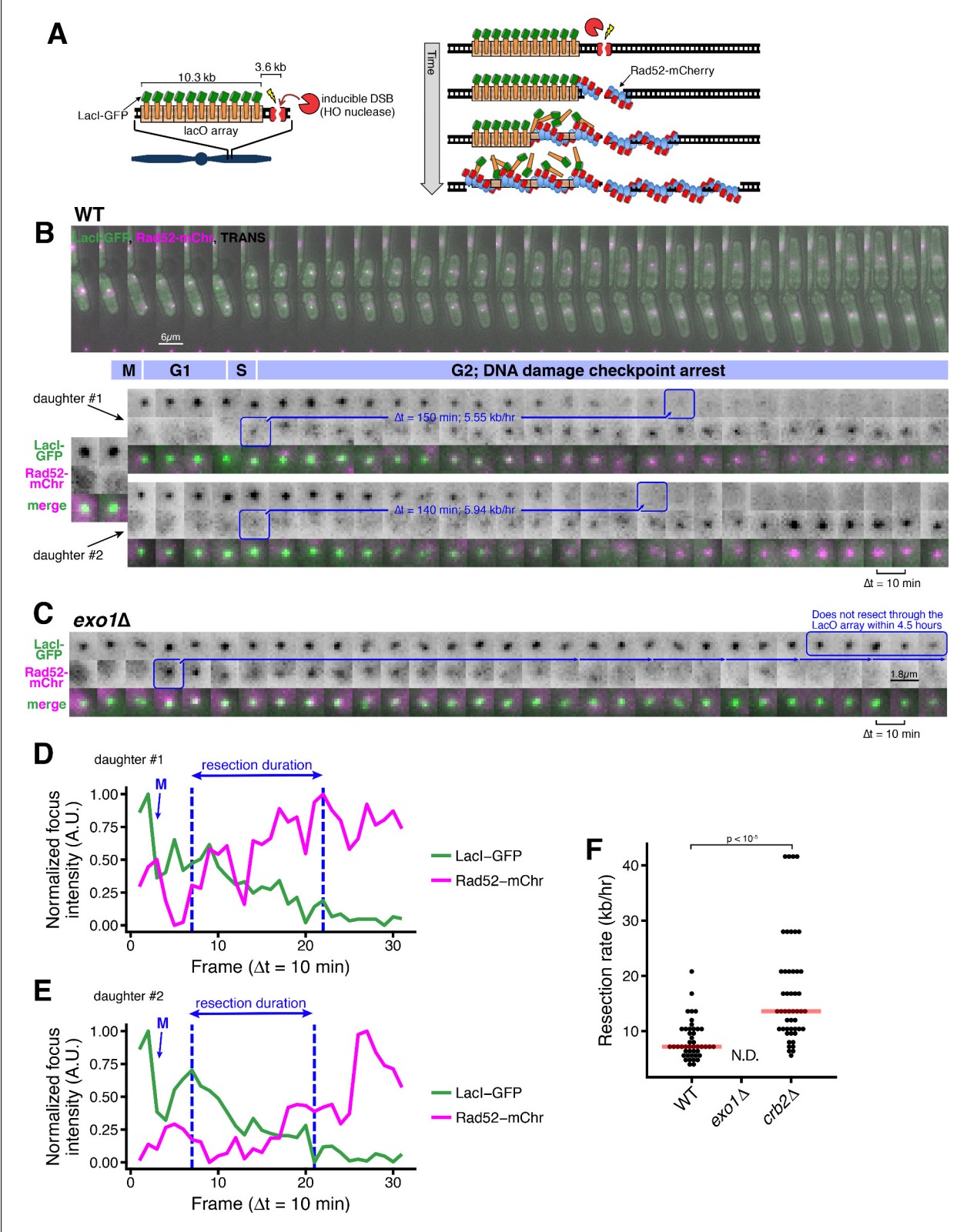

**Figure 1.** A microscopy-based assay to measure long-range resection in single cells. (**A**) Design of the LacO resection assay in *S. pombe*. HO endonuclease cut cite (HOcs) and LacO integration at the *Mmf1* locus on Chr II allows live-cell measurements of resection rates. Rad52-mCherry loads on DSB ends after resection initiation, and LacI-GFP is displaced as resection creates long tracts of ssDNA through the LacO array. (**B**) DSB resection events in two WT daughter cells. The majority of the *S. pombe* cell cycle is spent in G2, and all DSBs are observed in S/G2 based on the timing of

*Figure 1 continued on next page*

*Figure 1 continued*

mitosis and cell fission. Images shown are maximum intensity Z-projections acquired at 10-min time intervals. Blue annotations denote the starting point of resection (first frame with a detectable Rad52 focus, shortly after resection begins) and the end point (first frame with total loss of the LacO/LacI-GFP focus) of individual resection events. These start/end frames mark the total duration of resection through the 13.87 kb distance between the HOcs and the distal end of the repetitive LacO array and are used to compute resection rate (kb/hr) for individual cells. (C) Representative resection-deficient *exo1Δ* cell that that loads on-target Rad52-mCherry but does not lose the LacO/LacI-GFP focus. Because resection of the LacO array is too slow to be completed during the window of data acquisition, we are not able to quantify the rate of resection in *exo1Δ* cells. (D and E) Quantification of LacO/LacI-GFP focus and Rad52-mCherry focus intensities over time for the cells shown in (B). The arrow and 'M' show the time of mitotic division, which leads to a decrease in GFP intensity. Quantification used full Z-stack images (not maximum intensity projections shown in B) at subpixel resolution with background normalization. See Materials and methods for more details. (F) Single-cell measurements of resection rate using the LacO resection assay. Horizontal red bars mark the median resection rate for each genotype. *exo1Δ* rates cannot be determined (N.D.) because resection through the LacO array does not complete within 5 hr of data acquisition. p-values shown are from pairwise two-tailed t-tests, using a Bonferroni correction for multiple comparisons. Number of biological replicates and counts of analyzed cells can be found in *Supplementary file 2*.

DOI: https://doi.org/10.7554/eLife.33402.003

The following figure supplements are available for figure 1:

**Figure supplement 1.** Design and validation of the LacO and HO cut site integrations.

DOI: https://doi.org/10.7554/eLife.33402.004

**Figure supplement 2.** Additional examples of the LacO resection assay in individual WT cells.

DOI: https://doi.org/10.7554/eLife.33402.005

**Figure supplement 3.** Validation and characterization of the LacO resection assay.

DOI: https://doi.org/10.7554/eLife.33402.006

in WT *S. pombe*, with the Rqh1/Sgs1/BLM and Dna2 resection pathway playing a secondary role (*Langerak et al., 2011*). As expected, cells lacking Exo1 show a persistent LacO/LacI-GFP focus over many hours even after Rad52-mCherry loads (*Figure 1C*). Importantly, loss of Exo1 does not influence the induction of the site-specific DSB as measured by quantitative loss of a PCR product across the cut site (*Figure 1—figure supplement 3A*).

This assay only visualizes DSB foci after resection initiation by Ctp1/Sae2/CtIP and MRN/MRX (~100 nt of resection), as Rad52 loads at DSBs by exchanging with RPA on nascent ,resected ssDNA ends (*Jensen and Russell, 2016*; *Lisby et al., 2004*; *Ma et al., 2015*; *Mimitou and Symington, 2008*). Using quantitative imaging with calibration strains, we estimate that visualization of the Rad52-mCherry focus requires loading of ~30 copies (see Materials and methods), representing a Rad52 filament equivalent to at least 90 nt (but likely several hundred nts, see below) of resected ssDNA on each sister in G2 (*Gibb et al., 2014*; *Grimme et al., 2010*; *Kagawa et al., 2002*; *Singleton et al., 2002*; *Swartz et al., 2014*; *Wu and Pollard, 2005*). To test the frequency with which resection initiation leads to visible Rad52 foci in this assay, we compared the timing of Rad52-mCherry foci formation within the cell population to an independent measure of resection using a quantitative PCR (qPCR) assay in which resection protects from digestion at an ApoI cut site 168 nt downstream of the HO cut site (*Langerak et al., 2011*) (*Figure 1—figure supplement 3B*). Using this qPCR-based assay, we observe similar DSB induction frequency and kinetics for ApoI protection and Rad52 focus formation, with an apparent 30–60 min delay between ApoI protection (qPCR) and Rad52-mCherry loading, which likely represents the time required for RPA loading and exchange to Rad52 (*Lisby et al., 2004*) (*Figure 1—figure supplement 3C*). As the visibility of the LacO array is an important endpoint for this assay, we estimate that the LacI-GFP focus is visible down to a LacO array length of <500 bps under these conditions, based on our ability to robustly detect the focus from a 1 kb LacO array integrated into *S. pombe* (*Figure 1—figure supplement 3D–G*).

By measuring the time interval between Rad52-mCherry focus formation and LacO/LacI-GFP focus disappearance, we can determine the time required to resect through the full LacO array in individual cells (*Figure 1B*, *Figure 1—figure supplement 2A-D*, blue boxes). For example, for the two daughter cells in *Figure 1B* arising from cell division, the time interval from Rad52-mCherry loading until loss of the LacO/LacI-GFP focus is 150 min for the upper cell ('1'), and 140 min for the lower cell ('2'). The progressive loss of LacI-GFP intensity and gain in Rad52-mCherry intensity during the duration of resection is further illustrated by quantitative image analysis (*Figure 1E–F* and *Figure 1—figure supplement 3A'–B'*). As we know the genomic separation of the LacO array and the HO cut site (13.9 kb, *Figure 1—figure supplement 1*), we can calculate the resection rate from the

resection duration. For example, the calculated resection rates for the two cells in *Figure 1B* are very similar at 5.55 and 5.94 kb/hr. Across all WT cells, we detect a median, long-range resection rate of 7.6 kb/hr (*Figure 1F*). This rate is somewhat faster than the resection rates measured by previous population-based assays using qPCR in *S. pombe* (4 kb/hr) or Southern blot in *S. cerevisiae* (4.4 kb/hr) (*Langerak et al., 2011*; *Zhu et al., 2008*). As this assay isolates the process of long-range resection after Rad52 loading, one possibility is that resection rates that measure both resection initiation and long-range resection give rise to slower rates. Importantly, using the qPCR approach to compare resection upstream of the DSB (which contains the LacO array) and downstream of the DSB (which does not) demonstrates that any influence of the LacO array on resection rate is minor (*Figure 1—figure supplement 3H*).

As resection in the absence of Exo1 is very inefficient (*Figure 1C*), the rate cannot be determined using this assay. However, we can infer an upper bound of the long-range resection rate of ~2.8 kb/hr for *exo1Δ* cells. We also note a strong inhibition of resection (comparable to *ctp1Δ* cells) as close as 300 nts from the DSB in cells lacking Exo1 as detected by qPCR (*Figure 1—figure supplement 3I*); this correlates with a defect in Rad52-mCherry loading in the imaging-based resection assay (*Figure 1—figure supplement 3J*). This observation confirms that the extent of resection required to form visible Rad52 foci in this assay discussed above (~30 molecules; <300 nt) partially requires Exo1-dependent resection in addition to MRN/MRX- and Ctp1/Sae2/CtIP-dependent resection initiation (*Symington, 2016*), consistent with a study in budding yeast suggesting a specific requirement for Exo1 in the early phase of resection post-initiation (*Saad et al., 2014*).

53BP1 (in human cells) and its orthologue Rad9 (in budding yeast) repress resection initiation (*Chapman et al., 2012*; *Ferrari et al., 2015*; *Symington, 2016*); in budding yeast, loss of Rad9 also increases resection efficiency (*Bonetti et al., 2015*). Applying the live cell resection assay to fission yeast lacking the orthologous Crb2, we observe a strong increases the median rate of resection to (13.9 kb/hr), with some individual cells demonstrating very fast (~40 kb/hr) resection rates (*Figure 1F*). Thus, we can readily assess factors that positively and negatively influence long-range resection rate using this new LacO-based assay.

## Rev7 inhibits long-range resection

Next, we examined how loss of Rev7 influences long-range resection at DSBs during S/G2. The duration of LacO array resection is shorter in *rev7Δ* cells than in WT cells (80 min or less, *Figure 2A–B* and *Figure 2—figure supplement 1*). In the population, the median long-range resection rate for *rev7Δ* cells is similar to cells lacking Crb2 (10.4 kb/hr, *Figure 2C*). As Rev7 also functions with Rev3 as part of the polymerase ζ complex in translesion synthesis, we also confirmed that Rev3 does not affect resection (*Figure 2C*), consistent with previous data showing that repressing resection during HR is a distinct function of Rev7 (*Boersma et al., 2015*; *Rosenberg and Corbett, 2015*; *Xu et al., 2015*). Using the orthogonal qPCR approach (*Figure 1—figure supplement 3B*), we confirm that in *rev7Δ* cells, more chromosomes with DSBs have undergone 3 kb and 13 kb of resection than WT between 90 and 180 min after HO induction, respectively (*Figure 2D*). Again, loss of Rev7 has no influence on the rate of DSB induction (*Figure 2E*).

## The RecQ helicase, Rqh1, rather than Exo1, drives hyper-resection in the absence of Crb2 and Rev7

We next asked if Crb2/Rad9/53BP1 and Rev7 inhibit long-range resection through the Exo1 pathway, the RecQ helicase (Rqh1)-Dna2 pathway, or both (*Symington and Gautier, 2011*). Interestingly, we find that the rapid long-range resection rate observed in *crb2Δ* or *rev7Δ* cells is entirely Exo1-independent (*Figure 3A–C*, *Figure 3—figure supplement 1*). In stark contrast, the rapid rate of resection in a *rev7Δ* single mutant is entirely dependent on the presence of Rqh1 (*Figure 3A*). Consistent with these observations, we find that loss of Rev7 is able to rescue the severe growth defect of *exo1Δ* cells on rich media plates containing camptothecin, consistent with a derepression of Rqh1-dependent resection in the absence of a functional Exo1 pathway (*Figure 3D*). When considering only precisely determined resection events, the average long-range resection rate in *crb2Δ* cells is not statically less than that of *crb2Δrqh1Δ* cells when correcting for multiple comparisons (p=0.16) (*Figure 3A*). However, in many *crb2Δrqh1Δ* cells, resection durations extend beyond the timeframe

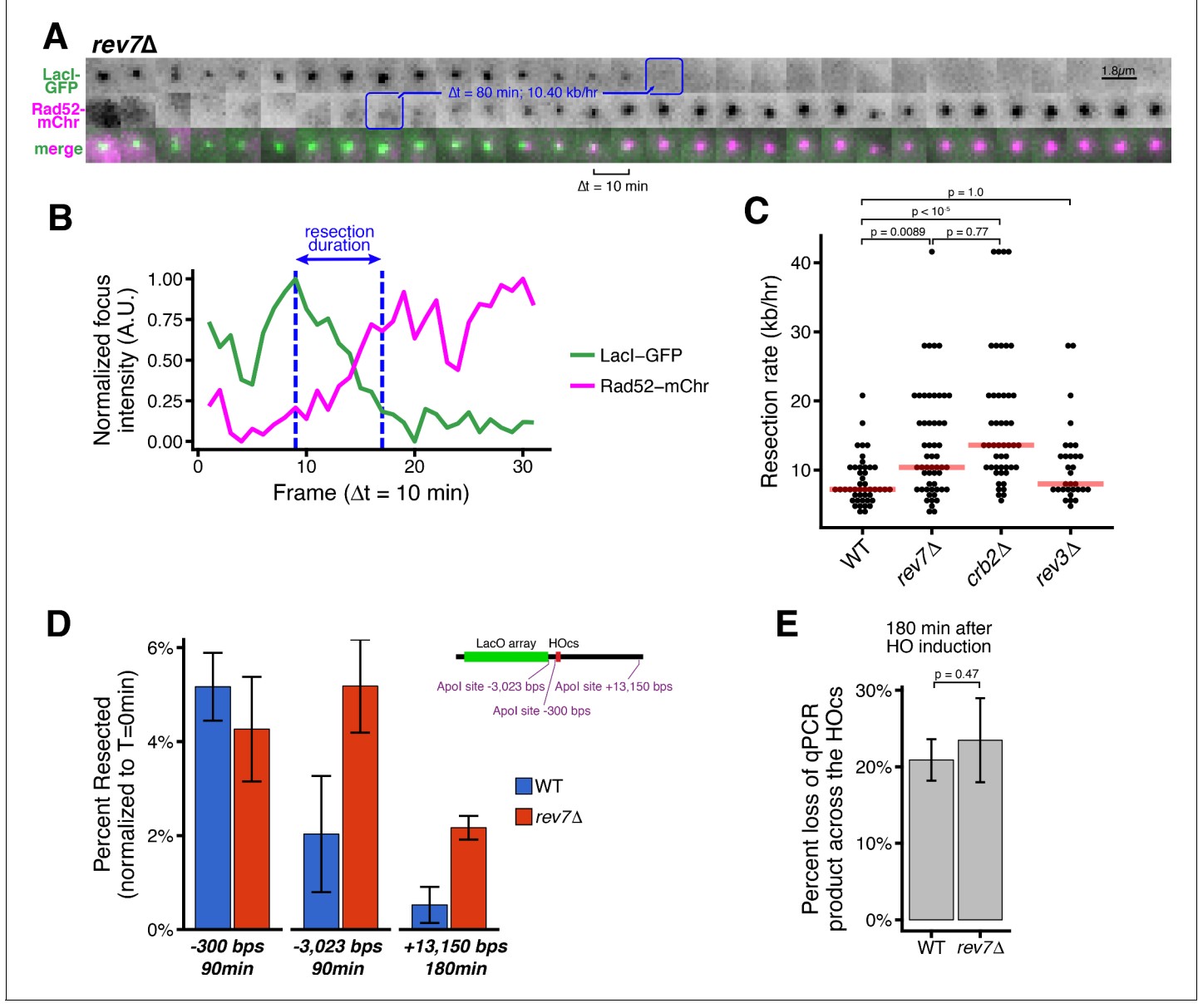

**Figure 2.** Loss of Rev7 causes an increase in long-rage resection comparable to the loss of Crb2. (A) Representative timeseries of resection through the LacO array in a single *rev7Δ* cell. Blue annotations mark the beginning and end of the resection event, as in *Figure 1B*. (B) Subpixel quantification of focus intensity for the *rev7Δ* cell resection event shown in (A). Quantification is as described in *Figure 1D,E* and in the Materials and methods. (C) Single-cell measurements of resection rate using the LacO resection assay. Horizontal red bars mark the median resection rate for each genotype. p-values shown are from pairwise two-tailed t-tests, using a Bonferroni correction for multiple comparisons. Number of biological replicates and counts of analyzed cells can be found in *Supplementary file 2*. (D) The long-range rate of resection in *rev7Δ* and WT cells, measured with an ApoI protection qPCR assay on a population level. ApoI cut site distances from the HOcs are indicated and shown in the diagram at right. Error bars show 95% CIs for at least three technical qPCR replicates across two or more biological replicates. (E) Similar to D, qPCR primers spanning the HOcs itself are used to monitor the efficiency of the HO cutting to form DSBs at 180 min after HO induction by uracil addition.

DOI: https://doi.org/10.7554/eLife.33402.007

The following figure supplement is available for figure 2:

**Figure supplement 1.** Additional examples of rapid long-range resection through the LacO array in *rev7Δ* cells.

DOI: https://doi.org/10.7554/eLife.33402.008

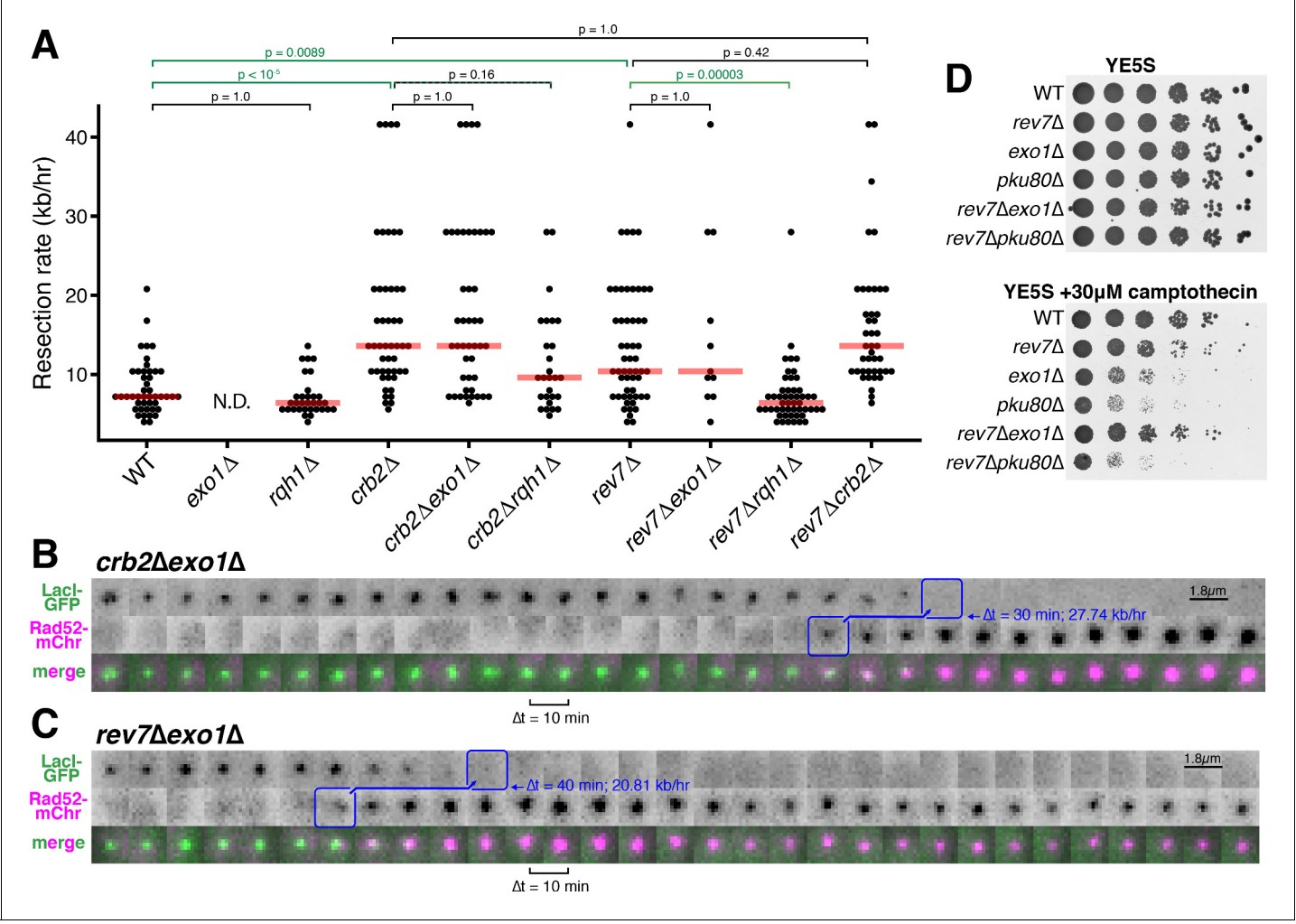

**Figure 3.** Rev7 and Crb2 act through the RecQ helicase, Rqh1, and not Exo1, to inhibit long-range resection. (**A**) Epistasis analyses of long-range resection rates from single-cell measurements. *exo1Δ* rates cannot be determined (N.D.) because resection through the LacO array does not complete within 5 hr of data acquisition. Red bars show median resection rates and p-values are from pairwise two-tailed t-tests, using a Bonferroni correction for multiple comparisons (significant comparisons shown in green). (**B,C**) Very rapid resection through the 10.3 kb LacO array is common in *crb2Δexo1Δ* (**B**) and *rev7Δexo1Δ* (**C**) cells, in contrast to *exo1Δ* single mutants that do not completely resect the LacO array within 5 hr (see *Figure 1C*). (**D**) Growth assay on rich media with and without camptothecin. Loss of Rev7 can rescue the severe growth defect of *exo1Δ* cells.

DOI: https://doi.org/10.7554/eLife.33402.009

The following figure supplements are available for figure 3:

**Figure supplement 1.** Additional examples of very rapid, Exo1-independent resection through the LacO array in the absence of Crb2 or Rev7.
DOI: https://doi.org/10.7554/eLife.33402.010

**Figure supplement 2.** Epistasis analysis including non-exactly determined resection events.
DOI: https://doi.org/10.7554/eLife.33402.011

of data acquisition, suggesting that rapid *crb2Δ* resection also requires Rqh1 (*Figure 3—figure supplement 2*).

Taken together, these results strongly suggest that loss of either Rev7 or Crb2 drives a shift in resection pathway mechanism from Exo1 to the RecQ helicase-dependent pathway. Unlike Exo1-driven resection that dominates in WT cells, *crb2Δ* or *rev7Δ* cells in which the RecQ helicase is derepressed are capable of resection rates in excess of 20 kb/hr, indicating that the RecQ helicases are more capable of driving hyper-resection of DSBs than the Exo1-dependent pathway. The strong inhibition of RecQ helicase-dependent resection by Crb2 and Rev7 also explains why Rqh1 is not a major player in WT *S. pombe* resection, since Rqh1 can be derepressed to such a large extent (by

loss of Crb2 or Rev7) that Exo1 becomes dispensable for long-range resection in *crb2Δexo1Δ* and *rev7Δexo1Δ* cells (*Figure 3A-C*, *Figure 3—figure supplement 1*).

## Discussion

Here, we demonstrate a microscopy-based assay capable of quantitatively measuring DSB end resection in living cells, specifically the long-range phase of resection catalyzed by Exo1 or Rqh1/Dna2. By allowing for the visualization of resection in single cells, this assay reveals individual long-range resection rates, which will provide access to information about the variability in resection efficiency within cell populations. Moreover, as this assay allows the tracking of the location of DSB lesions within the nucleus, it will be able to uniquely interrogate the role of intranuclear architecture on resection rates and pathways in the future.

By leveraging the advantages of this LacO-based assay, we show that Crb2/Rad9/53BP1 and Rev7 both act as specific inhibitors of RecQ helicase-mediated long-range resection of DSBs (*Figures 2* and *3*), supporting a model in which loss of these resection inhibitors drives a change in resection pathway rather than boosting Exo1-dependent resection. This is consistent with a previous study that identified specific mutations in the budding yeast RecQ helicase, Sgs1, that can disrupt inhibition by the Crb2 orthologue, Rad9, leading to a gain in Sgs1-dependent resection (*Bonetti et al., 2015*), as well as a recent study highlighting the ability of Rad9 to antagonize resection at stalled replication forks by repressing a Dna2-dependent pathway (*Villa et al., 2018*). Integrating these findings together with this study, we suggest that Crb2/Rad9 and Rev7 enforce Exo1-dependent long-range resection in yeasts, which prevents the hyper-resection that we find is characteristic of the RecQ helicase/Dna2 pathway (*Figure 3* and *Bonetti et al., 2015*). In support of this model, we find that loss of Rev7 is able to restore cell viability of *exo1Δ* cells on media containing camptothecin, suggesting that derepression of Rqh1 can substitute for loss of Exo1, consistent with the gain in resection rate in cells lacking both Rev7 and Exo1 (*Figure 3*).

The ability of Rev7 to antagonize RecQ helicase/Dna2-dependent long-range resection in fission yeast is very likely to be conserved in mammalian cells. Indeed, indirect evidence suggests that Rev7 knock-down promotes a gain in resection, as loss of Rev7 rescues CtIP-dependent RPA and Rad51 loading at irradiation-induced DSBs in cells lacking BRCA1 (*Xu et al., 2015*). Importantly, the resection pathway responsible for the gain in repair factor loading was not investigated in this study, but our work suggests that BLM (acting with DNA2) is a likely candidate. Loss of Rev7 does lead to longer 3' single-stranded G-rich overhangs at telomeres in cells with inactivated TRF2 (a component of the shelterin complex); in this case, only a partial rescue was obtained by co-depletion of CtIP or Exo1 (*Boersma et al., 2015*). Again, we hypothesize that this finding likely reflects contributions of BLM and DNA2, although it remains possible that the impact of Rev7 on long-range resection during G2 figures more prominently in fission yeast.

Critically, our findings suggest a new mechanism by which loss of either 53BP1 or Rev7 allows $BRCA1^{-/-}$ $p53^{-/-}$ cells to become resistant to PARP inhibitors (*Jaspers et al., 2013*; *Xu et al., 2015*). It has been proposed previously that the compromised CtIP- and Exo1-dependent resection in BRCA1-deficient cells can be restored by loss of 53BP1 or Rev7 (*Boersma et al., 2015*; *Xu et al., 2015*), thereby overcoming the increased DSB load caused by PARP inhibitors (*Polato et al., 2014*; *Tomimatsu et al., 2012*). Our study reveals that not just the efficiency, but also the molecular mechanism of resection is altered upon loss of Crb2/53BP1 or Rev7. We expect these insights to have several consequences. First, RecQ helicases, when paired with DNA2, are capable of exceptionally fast resection in vitro (*Niu et al., 2010*), which when derepressed by loss of 53BP1 or Rev7 could cause extended tracts of ssDNA several kb long, promoting further genome instability (*Hicks et al., 2010*; *Ochs et al., 2016*); indeed, our data point to such a hyper-resection phenotype upon loss of either Crb2 or Rev7 (*Figure 2C*). Second, our results suggest that inhibitors of RecQ helicases could potentially re-sensitize BRCA1-null cells to PARP inhibitors, as this would make them both Exo1- and RecQ helicase-deficient (*Aggarwal et al., 2013*; *Yazinski et al., 2017*).

# Materials and methods

## Cell culture, strain construction and verification

The strains used in this study are listed in *Supplementary file 1*. *S. pombe* were grown, maintained, and crossed using standard procedures and media (*Moreno et al., 1991*). Gene replacements were made by exchanging open reading frames with various MX6-based drug resistance genes (*Bähler et al., 1998*; *Hentges et al., 2005*). The 10.3 kb LacO array was inserted between *Mmf1* and *Apl1* on the right arm of chromosome II (Chr II: 3,442,981) using a modified two-step integration procedure that first creates a site-specific DSB to increase targeting efficiency of linearized plasmid pSR10_ura4_10.3kb (*Leland and King, 2014*; *Rohner et al., 2008*). A modified MX6-based hygromycin-resistance cassette containing the HO cut site was then inserted between *Apl1* and *Mug178* on chromosome II (Chr II: 3,446,249), 3.2 kb distal to the LacO insertion. The total distance between the HO cut site and the beginning of the 10.3 kb LacO array is 3.57 kb. As the LacO array can contract during the process of transformation, integrants were screened by HincII digest followed by southern blot (*Figure 1—figure supplement 1*) using standard procedures, a biotin-conjugated LacO probe, and a streptavidin-HRP chemilluminescent detection system (Thermo #N100 and #34096).

## DSB induction using $P_{urg1lox}$-HO

We used the uracil-responsive $P_{urg1lox}$ expression system, with slight modifications, to induce HO endonuclease expression and create site-specific DSBs at the HO cut site (*Watson et al., 2011*; *Watt et al., 2008*). We performed a fresh integration of the *HO* gene at the endogenous *urg1* locus for each experiment in order to reduce long-term instability at the HO cut site or the development of HO resistance, presumably due to insertion/deletion events caused by basal expression levels of HO. The pAW8E*Nde*I-HO plasmid (a gift from Tony Carr) was transformed into *S. pombe*, which were then plated onto EMM-leu+thi-ura plates (-leucine: plasmid selection; +thiamine: $P_{nmt1}$-Cre repression; -uracil: $P_{urg1lox}$-HO repression). After 4–5 days of growth at 30°C, 40–100 individual colonies were combined to obtain a reproducible plasmid copy number across the population. Cre-mediate *HO* gene exchange at the endogenous Urg1 locus (*urg1::RMCE$_{bleMX6}$*) was induced by overnight culture in EMM-thi-ura+ade+NPG media (-thiamine: expression of Cre from pAW8E*Nde*I-HO; -uracil: $P_{urg1lox}$-HO repression; +0.25 mg/mL adenine: reduce autofluorescence; +0.1 mM n-Propyl Gallate (NPG): reduce photobleaching in microscopy experiments, prepared fresh). The following day, site-specific DSBs were induced in log-phase cultures by the addition of 0.50 mg/mL uracil. This induction strategy resulted in ~15% of cells making a DSB within ~2 hr (*Figure 1—figure supplement 3C*).

## qPCR resection assay

Initiation of resection was assessed using a previously described qPCR assay where ssDNA produced by resection causes protection from ApoI digestion (*Langerak et al., 2011*). ApoI cut site positions (relative to the HO cut cite (Chr II: 3446192)) and PCR primer sets spanning each ApoI recognition site can be found in *Supplementary file 3*. Mock HincII digestions (do not affect qPCR products) and additional control primers at Ncb2 were used to normalize for ApoI digestion efficiency (see *Supplementary File 3*).

## Microscopy

All images were acquired on a DeltaVision widefield microscope (Applied Precision/GE) using a 1.2 NA 100x objective (Olympus), solid-state illumination, and an Evolve 512 EMCCD camera (Photometrics). Slides were prepared ~20 min after adding 0.50 mg/ml uracil to log-phase cultures to induce HO endonuclease expression and DSB formation. Cells were mounted on 1.2% agar pads (EMM +0.50 mg/mL uracil, +2.5 mg/ml adenine, +0.1 mM freshly prepared NPG) and sealed with VALAP (1:1:1 vaseline:lanolin:paraffin). Image acquisition began between 40 and 80 min after uracil addition. Imaging parameters for all resection assay data acquisition were as follows. Transmitted light: 35% transmittance, 0.015 s exposure; mCherry: 32% power, 0.08 s exposure; GFP: 10% power, 0.05 s exposure. At each time point (every 10 min for 5–7 hr), 25 Z-sections were acquired at 0.26 μm spacing. Identical imaging parameters were used to image a strain expressing endogenously tagged Sad1-mCherry (Sad1 forms a single focus at the spindle pole body that contains between 450 and 1030 molecules) and relative mCherry foci intensities were used to determine that ~30

molecules of Rad52-mCherry are required to detect a visible focus with these imaging parameters (*Wu and Pollard, 2005*).

## Image analysis

For the LacO resection assay, every cell cycle was tracked and quantified individually, including timing of nuclear division, cellular division, Rad52-mCherry focus formation, and LacO/LacI-GFP focus disappearance. Only on-target Rad52 foci (that co-localized with LacO/LacI-GFP for at least 2 frames) were considered, since many DSB events occur throughout the genome spontaneously, especially during S-phase. The number of cells and events used to generate the plots in all Figures is included as *Supplementary file 2*. The time between the first frame with an on-target Rad52-mCherry focus and the first frame with complete disappearance of the LacO/LacI-GFP focus is the duration of resection through 3.57 kb (between the HO cut site and the start of the LacO repeats) plus the full 10.3 kb LacO array. All fields from all genotypes were input into custom ImageJ macros that randomized the order of the fields/genotypes, blinded the images by removal of the file names, set the contrast to be identical for every image, and numbered each cell lineage. Each blinded field was then manually assessed for photobleaching of the LacO/LacI-GFP foci in cells without induced DSBs (>80% of all cells) to ensure that disappearance of any LacO/LacI-GFP foci in cells with on-target DSBs was due to resection through the LacO array rather than photobleaching of the GFP signal. Next, using the pre-determined contrast settings for mCherry and GFP channels (to maintain consistency across all images analyzed) individual cells which had on-target DSB events were manually identified, and scored for the first frame of Rad52-mCherry focus appearance and then the first frame in which the LacO/LacI-GFP focus had completely disappeared.

For focus intensity plots (e.g. *Figure 1D,E*), quantification was performed in ImageJ on the full 5D image stacks (not maximum intensity projections, which are shown in the image panels throughout for ease of viewing, for example *Figure 1B*). Subpixel measurements were made in a cylinder approximating the point spread function surrounding the manually scored subpixel center of the focus. A cylindrical shell surrounding the focus was used for background subtraction in both the GFP and mCherry channels.

Raw data were processed, visualized, and analyzed using R, in particular packages dplyr, ggplot2, and broom. Raw data, raw analysis for all individual cells included in plots, complete code, and other supporting materials are publically available on GitHub https://github.com/lelandbr/Leland_King_2018_eLife_Rev7_EndResection (*King and Leland, 2018*; copy archived at https://github.com/elifesciences-publications/Leland_King_2018_eLife_Rev7_EndResection).

## Growth assays

Cells were grown overnight in YE5S media. Concentrations for each culture were monitored by both $OD_{600}$ and a Coulter Principle cell counter (Orflow Moxi Z). Cultures were diluted as needed to ensure identical numbers of cells were spotted for each genotype, starting with ~$4 \times 10^6$ cell/mL and going down by sixfold dilutions. Plates were prepared using standard procedures (*Moreno et al., 1991*), with the addition of 30 µM camptothecin (Sigma; ≥95% HPLC purified) after autoclaving.

## Acknowledgements

We would thank the Drs. Susan Gasser, Tony Carr, Paul Russell, Li-Lin Du, Masayuki Yamamoto, and Julia Cooper for strains and plasmids; Jessica Johnston for generating and imaging the 1 kb LacO array strain; and Topher Carroll and Tom Pollard for helpful input and feedback. This work was supported by a National Science Foundation Graduate Research Fellowship (DGE-1122492), NIH training grant T32-GM007223, and a Gruber Science Fellowship to BAL; and the Searle Scholars Program, a Pilot Grant from the Yale Cancer Center, and the National Institutes of Health Office of the Director (DP2OD008429-01) to MCK.

## Additional information

### Funding

| Funder | Grant reference number | Author |
| --- | --- | --- |
| National Science Foundation | DGE-1122492 | Bryan A Leland |
| The Gruber Foundation | Gruber Science Fellowship | Bryan A Leland |
| National Institutes of Health | T32-GM007223 | Bryan A Leland |
| National Institutes of Health | DP2OD008429-01 | Megan C King |
| Searle Scholars Program | Scholar Award | Megan C King |
| Yale Cancer Center | Pilot Grant | Megan C King |

The funders had no role in study design, data collection and interpretation, or the decision to submit the work for publication.

### Author contributions

Bryan A Leland, Conceptualization, Data curation, Software, Formal analysis, Funding acquisition, Validation, Investigation, Visualization, Methodology, Writing—original draft, Writing—review and editing; Angela C Chen, Data curation, Formal analysis, Investigation, Visualization; Amy Y Zhao, Formal analysis, Validation, Investigation, Visualization; Robert C Wharton, Validation, Investigation, Methodology; Megan C King, Conceptualization, Supervision, Funding acquisition, Investigation, Methodology, Writing—original draft, Project administration, Writing—review and editing

### Author ORCIDs

Megan C King https://orcid.org/0000-0002-1688-2226

### Decision letter and Author response

Decision letter https://doi.org/10.7554/eLife.33402.021
Author response https://doi.org/10.7554/eLife.33402.022

## Additional files

### Supplementary files

• Supplementary file 1. Strains used in this study. These strains were for all LacO resection assay experiments in *Figures 1–3* unless otherwise noted. $P_{Dis1}$-GFP-LacI-NLS is derived from *Shimada et al. (2003)*. LacO integrations were performed as described in *Leland and King, 2014*. Strains containing the RMCE $P_{urg1lox}$ expression system were derived from *Watson et al. (2011)*.
DOI: https://doi.org/10.7554/eLife.33402.012

• Supplementary file 2. Number of biological replicates, cell cycles, and DSB events analyzed for each genotype. These counts pertain to all data from the resection assay in *Figures 1–3*.
DOI: https://doi.org/10.7554/eLife.33402.013

• Supplementary file 3. Primers sets used in qPCR measurements of resection.
DOI: https://doi.org/10.7554/eLife.33402.014

• Transparent reporting form
DOI: https://doi.org/10.7554/eLife.33402.015

### Data availability

Raw analysis for all individual cells included in plots, complete code, and other supporting materials are publicly available on GitHub github.com/lelandbr/Leland_King_2018_eLife_Rev7_EndResection. The raw movies for representative cells presented in the figures have been uploaded to Dryad [doi:10.5061/dryad.1db5500]. The full raw datasets (all cells, all fields, all movies) are available on request from the corresponding author (megan.king@yale.edu) as they are TBs in size.

The following dataset was generated:

| Author(s) | Year | Dataset title | Dataset URL | Database, license, and accessibility information |
|---|---|---|---|---|
| Bryan A Leland, Megan C King | 2018 | Rev7 and 53BP1/Crb2 prevent RecQ helicase-dependent hyper-resection of DNA double-strand breaks | http://dx.doi.org/10.5061/dryad.1db5500 | Available at Dryad Digital Repository under a CC0 Public Domain Dedication |

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
