## [Decision Letter]

[Editors’ note: a previous version of this study was rejected after peer review, but the authors submitted for reconsideration. The first decision letter after peer review is shown below.]

Thank you for submitting your work entitled "Rev7 and 53BP1/Crb2 prevent RecQ helicase-dependent hyper-resection of DNA double-strand breaks" for consideration by *eLife*. Your article has been reviewed by three peer reviewers, one of whom is a member of our Board of Reviewing Editors and the evaluation has been overseen by a Senior Editor. The reviewers have opted to remain anonymous.

Our decision has been reached after consultation between the reviewers. Based on these discussions and the individual reviews below, we regret to inform you that your work will not be considered further for publication in *eLife*.

The overall findings with regard to Crb2 and Rev7 inhibiting the Rqh1 pathway are interesting, but there were many concerns with how the data are collected and analyzed. The manuscript presents novel data based on an assay that is not fully substantiated. Many controls are still needed to support the validity of the underlying assay.

There were unanimous reservations and many discussions about the assay which, in turn, undermined the confidence in, and significance of, the conclusions. Although most of the conclusion were consistent with existing facts, there were some specific inconsistencies noted by the reviewers. Overall, the results seem incomplete or preliminary.

Essential revisions:

Reviewer #1:

This is an interesting paper with potentially important conclusions. The results depend entirely on the assay, which seems to be reliable, but some questions remain:

1) There are no graphs showing the quality of the kinetic data. The authors need to show graphs of intensity vs time, for each assay, with fitting statistics.

2) It's a bit surprising that the array of 256 copies of lac repressor protein doesn't affect the measured rate. The authors need to show overlaid graphs of intensity vs time for the arrays; the PCR data for the arrays with lac repressor; and PCR data for the DNA arrays without lac repressors.

3) In the experiments with tagged RAD52, I don't understand why the foci persist. I would have expected RAD51 to replace the RAD52. Although the intensity does decrease (maybe just from photobleaching), it seems to be slower than the observations from the Rothstein lab on Rad52 in *S. cerevisiae*.

As stated above, the results are interesting and informative. However, the results rely entirely on the validity of the assay. In the current version, the authors don't present enough analysis and comparisons of the kinetic data to convincingly establish the assay. Presumably, they have the data; they need to provide it.

*Reviewer #2:*

DNA end resection is essential for homologous recombination, but excessive end resection can be detrimental to genome integrity. Long-range resection is catalyzed by either Exo1 or by a RecQ family helicase in collaboration with Dna2. The relative contribution of these two mechanisms and how they are regulated is not well understood. Here, the authors use a cytological approach to study end resection in single cells. An HO cut site was inserted 3.4-kb from a 10.3-kb lacO array located on Ch 2 of *S. pombe*. DSB formation/early resection was detected by the appearance of Rad52-mCherry foci that co-localize with the lacO array marked with LacI-GFP. Resection of the entire lacO array results in loss of the GFP signal while the mCherry signal remains. Consistent with a previous study using population based qPCR to measure resection (Langerak et al., 2011), the authors report that most resection is due to Exo1 activity and not Rqh1. Two years ago, several groups reported that Rev7 acts with 53BP1 to inhibit resection in mammalian cells. Here, the authors show that Rev7 and Crb2/53BP1 have a conserved role in preventing long-range resection in *S. pombe*. Furthermore, they show that Rev7 and Crb2 specifically block resection by Rqh1.

Overall, the authors demonstrate that live cell imaging can be used to study end resection in single cells, and their studies show considerable cell-to-cell variation in the initiation of resection, a feature missed by population-based DNA analysis. However, I'm not convinced that the microscopy-based single-cell assay is the best way to monitor resection. The generated data imply that resection speed can be measured accurately but that relies on some assumptions. In subsection “Image analysis” the authors write "The time between the first frame with an on-target Rad52-mCherry focus and the first frame with complete disappearance of the LacO/LacI-GFP focus is the duration of resection through 3.57 kb (between the HO cut site and the start of the LacO repeats) plus the full 10.3 kb LacO array". Is resection through the complete LacO array necessary for disappearance of the GFP focus? I could imagine that the GFP focus disappears before the complete LacO array is degraded. What is the minimum number of detectable LacO repeats? The authors could integrate LacO arrays of different lengths and check what the detection limit is. Or they should at least confirm the timing of the GFP focus disappearance with the qPCR assay (According to Figure 1—figure supplement 1 they could use HincII for this).

Also, to my eye the Rad52 focus appearance and LacI-GFP focus disappearance are not easily identified (based on the microscopy images shown). The authors should mark the frames they define as "Rad52 focus appearance" and "LacI-GFP focus disappearance". In the Materials and methods section, they don't say how they define these events. Do they do it manually or using some image analysis software? What is the threshold? More details are necessary here to judge the accuracy of the method. Also, the restriction of the assay to 5 hours by photobleaching seems to be a considerable drawback, which limits the number of "usable" cell trajectories (as shown in Figure 2—figure supplement 2B). Perhaps by changing the imaging period and number of z-stacks they could extend the time window for monitoring resection. Other approaches, such as using more photostable fluorescent proteins (e.g. LacI-mKate2, Rad52-monomeric NeonGreen), LEDs as light sources, or switching to an enzyme that cuts more efficiently than HO (I-PpoI has been successfully used in *S. pombe* and the efficiency of cutting is better than HO) would involve considerable more effort and time.

The authors stress that they use a single cell assay. But it is not clear what the advantage of single cell data is in their work. In the end they are comparing population medians, which could also be generated with bulk experiments. The low DSB formation efficiency might be a motivation to use a single cell assay to restrict the analysis to the few cells with a DSB. However, bulk experiments generally take the cutting efficiency into account, e.g. the qPCR-based assay described by Zierhut and Diffley, (2008) considers the HO cut fraction.

Rad52 foci are used as a read out for DSB formation and initiation of resection. Can these two steps be separated, for example, by measuring DSB formation by qPCR using primers flanking the HO cut site on the same samples used for the ApoI protection assay, or possibly using Mre11-mCherry instead of Rad52? It appears from Figure 1B (upper panel) that resection through most of the lacO array is required for a strong Rad52 signal.

In theory, the HO-induced DSB is unrepairable, but because the HO cutting efficiency is quite low, and *S. pombe* cells are in G2 most of the time, if one sister chromatid was cut and engaged in repair with the uncut sister it could result in an underestimation of resection. Were cells with a transient Rad52 focus detected? The exo1 example in Figure 1 appears to have a Rad52 focus that appears early and then goes away. Does elimination of Rad51 change the number of cells resecting or rate of resection? At late times, when the mCherry signal is very bright, are both sister chromatids cut and resected?

Langerak et al., used a qPCR assay to measure end resection and reported no defect in resection initiation (35 nt from HO cut site) in the exo1 mutant or exo1 rqh1 double mutant. Is the failure to detect Rad52 foci in most exo1 cells because resection tracts are <90 nt or because the amount of ssDNA required to support a Rad52 focus is much longer than 90 nt? Given that the Rad52 single is quite weak until the lacO array disappears, I think the authors might be under-estimating the amount of ssDNA to visualize a Rad52 focus. Does the exo1 mutant show normal DSB formation (measured with primers flanking the HOcs) and resection to the ApoI site located 168 nt from HOcs? Similarly, is DSB formation normal in the rev7 mutant and can early resection be detected by the qPCR assay?

Why does rqh1 suppress the early resection defect of rev7? An odd result that is not discussed in the text.

While the overall findings on the role of Rev7 and Crb2 repressing the Rqh1 resection pathway are certainly of interest to the field, the data analysis needs to be improved.

Reviewer #3:

DNA end resection is a process that initiates recombination-based DNA double strand break repair. As resection also generally inhibits non-homologous end-joining, regulation of DNA end resection is an important process. The human 53BP1 protein, though its various effectors, has been found to be an inhibitor of DNA end resection, although mechanistic insights are lacking. These processes appear to be at least partially conserved in low eukaryotes.

The authors are using a *S. pombe* as a model system, where they developed an assay allowing the monitoring of resection in live cells (Figure 1). The assay is based on the disappearance of GFP-LacI and appearance of Rad52-mCherry signal next to HO-endonuclease induced DSB. This is an interesting method that will be useful for high throughput microscopy-based screenings. However, the method has certain limitations in contrast to established southern blotting, PFGE or RTPCR-based methods:

a) The assay measures Rad52 accumulation, which is a step after resection. Rad52 is a mediator that loads Rhp51 on RPA-coated resected DNA. Using RT-PCR based assay, the authors established that there is a correlation between both processes (resection and RAD52 loading). However, this was only done in wt background, and it is possible RAD52 loading might differ in the mutants analyzed, and Rad52 might be loaded with different kinetics dependent on the resection pathway. The authors calculate resection rates making the assumption that there is no difference.

b) The assay measures resection of DNA bound by GFP-LacI, a non-physiologic binder. Therefore, the resection proteins must displace LacI for resection to occur. Under physiologic conditions, DNA near DSBs will likely be chromatinized and subsequently remodeled allowing resection. This is especially a concern when analyzing the role of chromatin binders such as 53BP1/Crb2. Also, the individual resection pathways may be affect by LacI binder to a different degree, which complicates interpretation.

The most interesting finding is that Rev7 and 53BP1/Crb2 appear to repress long-range resection dependent on Rqh1 (Figure 2): In the absence of Rev7 and Crb2, long-range resection is accelerated, which is independent of Exo1 and depends on Rqh1.

1) This observation should be verified using a previously established assay, given the concerns listed above. Related to this point, it remains a formal possibility that the action of Rqh1 is not specific to resection, but to strip LacI, allowing resection through another process. Performing resection with the same mutants in an established setup (with no LacI) will address this concern as well.

The authors then go on to demonstrate differential effects of Rev7 and Crb2 on early resection steps, based on timing of Rad52 foci appearance upon break induction (Figure 3). I am concerned about these results: using Rad52 (a protein with a function downstream of resection) does not appear to be correct as a marker of "early" resection.

2) The results should be analyzed in an established assay where the readout is clearly early resection. Also, to make claims on early resection, the analysis should include mutants deficient in short-range resection (e.g. mre11) and mutants in long-range resection, where both pathways have been inactivated (e.g. exo1 rqh1).

3) In Figure 2, the authors demonstrate that rev7 mutants have accelerated long-range resection, which is dramatically decreased when rqh1 is additionally mutated (rev7 rqh1), supporting the hypothesis that resection in rev7 mutants is Rqh1 dependent. In contrast however, the in Figure 3, the "early" resection is increased when rqh1 is mutated in rev7 background, a completely opposite effect. This is very confusing. The authors comment that "early" resection is a combination of Mre11 and Exo1/Rqh1 dependent processes. This dichotomy reinforces my concerns about the robustness of the experimental setup.

In summary, the manuscript presents an interesting assay and interesting pieces of data, but it seems rather preliminary at this point.

[Editors’ note: what now follows is the decision letter after the authors submitted for further consideration.]

Thank you for submitting your article "Rev7 and 53BP1/Crb2 prevent RecQ helicase-dependent hyper-resection of DNA double-strand breaks" for consideration by *eLife*. Your article has been reviewed by two peer reviewers, and the evaluation has been overseen by a Reviewing Editor and Kevin Struhl as the Senior Editor.

The reviewers have discussed the reviews with one another and the Reviewing Editor has drafted this decision to help you prepare a revised submission.

The biological findings of this manuscript are interesting, particularly the role of Rev7 in suppressing resection by the Rqh1 pathway, but there are significant concerns over the reliability of the assays used. The authors place equal importance on the fluorescence-based resection presented; however, the reliability of the assay is questionable and the weak point of this work. The conceptual idea of the assay is good, but there are reservations about its implementation and use to quantify resection speed. The authors have, for the most part, established the utility of their assay for the analysis of long-range resection, although they still need to provide a graphical example of their quantification. They discovered interesting roles of Rev7 and Crb2 in Rqh1-dependent resection. As both a methods contribution and a contribution to the recombination field, these findings could justify publication of a revised manuscript.

However, the manuscript then goes on to dissect the roles of these proteins in proximal recombination. Here the work is both incomplete and internally inconsistent. It would appear that the data in Figure 4A and Figure 4C are inconsistent with one another: 4A show a reduction in "Cell cycles with Rad52 foci" for rev7delta, whereas 4C shows no change, within error, for rev7delta at 300 bps (The figure legend, as well as the associated text in the main body of the manuscript, is not justified: "The extent of resection 300 bps and 3 kb from the HO cut site as assessed by the restriction enzyme/qPCR method supports less efficient resection initiation in rev7Δ cells compared to WT at 300 bp…").

Also, the conclusion that of Rev7 promotes early resection is based entirely on how quickly a Rad52 focus forms after break induction, and how many cells have visible Rad52 foci. It was suggested that the authors use a more direct measurement of break induction by qPCR using primers flanking the HO cut site. This has not been done and is a significant concern. If one compares the qPCR data for resection 168-bp from the HO cut site shown in Figure 1D and Figure 1—figure supplement 3B there are vast differences – 3% resection at 120 min in one figure compared with 15% resection at 120 min in the other. If there is this much variability between populations of cells then it could explain why the rev7 mutant looks different to wild type, and why rqh1 appears to suppress the early resection defect of rev7. How many independent inductions were performed for image analysis? The authors need a reliable method to assess DSB formation independent of Rad52 focus formation before drawing conclusions about a role for Rev7 in promoting early resection.

Furthermore, In Figure 1D, the authors use a qPCR assay to detect formation of ssDNA 168 and 14,253 bp from the HO cut site. From this assay, very few (2%) cells exhibit resection to the end of the lacO array 360 min after HO induction. This would appear to contradict the microscopy assay, which shows resection through the array takes ~150 minutes. Also, data should really be from biological replicas, not technical replicates of qPCR.

Consequently, whether or not Rev7 has a role in proximal resection is not clear from the authors' data.

There was agreement that manuscript needs to be revised:

1) It is essential that the confusing data on short range resection are clarified or removed from the manuscript. If the proximal resection data are retained, then the comments raised above need to be addressed. In addition, in the review of the prior version of this manuscript, reviewer #3 commented on this part in the original submission: "Also, to make claims on early resection, the analysis should include mutants deficient in short-range resection (e.g. mre11) and mutants in long-range resection, where both pathways have been inactivated (e.g. exo1 rqh1)." The authors did not address this previous request for clarification of this unexpected finding in this revised manuscript, using the analyses described for proximal resection (i.e., Figure 4).

It is unclear whether the authors can make these revision within the timeframe given. If not, then these data on proximal resection would need to be removed, and the conclusions of the manuscript refocused. One major finding is the effect of Crb2 on long-range resection: while this is new in *S. pombe*, it is well-described in *S. cerevisiae* (Rad9). The second major finding is the effect of Rev7 on long-range resection: this was only shown in humans, but not in the microbial eukaryotes. Although significant, the impact of the current manuscript is diminished without the proximal resection data; consequently, the author would need to make the contributions of their sound work much clearer in an expanded Discussion section of their work in relation to the existing literature (in any event, the existing Discussion section is inadequate.).

2) The authors use appearance of a Rad52-mCherry focus to identify onset of resection and disappearance of a lacO/LacI-GFP focus to identify resection past a site some 14 kb away from the DSB site. The problem is that both signals are rather fuzzy in many of the image series shown. This is especially true for the Rad52-mCherry focus (see e.g. the lower image series in Figure 1B, the first three image series in Figure 1—figure supplement 2, Figure 2A). The decision if a signal is judged as a focus or not is crucial, as it is the basis to calculate resection speed. This decision is made manually. Although the authors try to "equalize" the error by randomizing the image series prior to analysis, it seems to be a quite ambiguous approach of questionable reliability. What is their criterion to judge if a focus is present or not? Do they compare with background control strains? Why don't the authors use software to quantify the fluorescence signals and generate intensity trajectories? There are several non-commercial image analysis software packages available dedicated to exactly this purpose. Based on thresholds defined by appropriate control strains, appearance and disappearance of the signals could then be identified in a more controlled and rational way. A minimal requirement is that the author provides an x-y graph of foci fluorescent intensity vs time for each their examples of time-lapse video data in the manuscript.

If the authors submit another revised version, then the decision to accept or reject will be final, and no subsequent revisions will be considered.

---

## [Author Response]

[Editors’ note: the author responses to the first round of peer review follow.]

Essential revisions:Reviewer #1:This is an interesting paper with potentially important conclusions. The results depend entirely on the assay, which seems to be reliable, but some questions remain:

We are pleased that the reviewer found our study interesting and have endeavored to support our conclusions with additional experimental data as well as some clarifications.

1) There are no graphs showing the quality of the kinetic data. The authors need to show graphs of intensity vs time, for each assay, with fitting statistics.

We agree that if the resection rates were derived by measuring the kinetic loss of GFPLacI intensity this would be a critical point. Indeed, this is an aspect of this approach that can be exploited. However, for the data reported in this manuscript, we found that the most robust approach to determining the median long-range resection rate is an endpoint measurement (see Materials and methods section for details; we now indicate the explicit time points in each time lapse example with blue boxes, alongside the resection duration and determined rate). Thus, we derive a rate from the resection of the entire LacO array rather than from the rate of GFP-LacI loss. This being said, as we show in Author response image 1, the difference in slopes for the rate of loss of GFP-LacI for individual cells in consistent with variability in the population that we describe quantitating based on the end point approach.

2) It's a bit surprising that the array of 256 copies of lac repressor protein doesn't affect the measured rate. The authors need to show overlaid graphs of intensity vs time for the arrays; the PCR data for the arrays with lac repressor; and PCR data for the DNA arrays without lac repressors.

To further verify the (perhaps surprising) result that the LacO array does not strongly influence the measured resection rates, we carried out qPCR analysis of resection within a population of WT cells on both sides of the induced DSB. Although we cannot use qPCR to assess resection within the LacO array because of its repetitive nature, we did assess the extent of resection on the far side of the LacO array (~14 kb from the DSB site) and compared this to resection ~14 kb on the other (non-LacO containing) side of the DSB. We find that the rates on both sides of the induced DSB appear to be highly similar (Figure 1D). Thus, although it may be surprising, we have not found any indication that the LacO array alters the process of resection. Further, our extensive analysis by qPCR in the revised manuscript is entirely consistent with the interpretations made from the LacO-based system we have developed.

3) In the experiments with tagged RAD52, I don't understand why the foci persist. I would have expected RAD51 to replace the RAD52. Although the intensity does decrease (maybe just from photobleaching), it seems to be slower than the observations from the Rothstein lab on Rad52 in S. cerevisiae.

To further verify the (perhaps surprising) result that the LacO array does not strongly influence the measured resection rates, we carried out qPCR analysis of resection within a population of WT cells on both sides of the induced DSB. Although we cannot use qPCR to assess resection within the LacO array because of its repetitive nature, we did assess the extent of resection on the far side of the LacO array (~14 kb from the DSB site) and compared this to resection ~14 kb on the other (non-LacO containing) side of the DSB. We find that the rates on both sides of the induced DSB appear to be highly similar (Figure 1D). Thus, although it may be surprising, we have not found any indication that the LacO array alters the process of resection. Further, our extensive analysis by qPCR in the revised manuscript is entirely consistent with the interpretations made from the LacO-based system we have developed.

As stated above, the results are interesting and informative. However, the results rely entirely on the validity of the assay. In the current version, the authors don't present enough analysis and comparisons of the kinetic data to convincingly establish the assay. Presumably, they have the data; they need to provide it.

Indeed, as suggested by the reviewer, we are pleased to be able to provide additional support for the interpretation for the vast majority of our assertions in the initial submission. In particular, we have carried out extensive qPCR analyses that support the fundamental soundness of the assay design and implementation (Figure 1D, Figure 1—figure supplement 3, Figure 2C, Figure 4C). However, as we argued in our initial submission, this single cell assay can reveal attributes of individual events that are poorly captured in the population-based qPCR data, particularly for cells lacking Rev7, in which slow initial steps of resection initiation and faster long-range resection are conflated in a population-based approach (see Figure 4C and the associates text). Indeed, we believe this is one of the reasons that this assay can be a powerful resource to complement existing population-based assays. We are confident that the inclusion of these additional data will provide the necessary validation to highlight the utility of this experimental approach and the resulting insights into DSB end resection.

Reviewer #2:DNA end resection is essential for homologous recombination, but excessive end resection can be detrimental to genome integrity. Long-range resection is catalyzed by either Exo1 or by a RecQ family helicase in collaboration with Dna2. The relative contribution of these two mechanisms and how they are regulated is not well understood. Here, the authors use a cytological approach to study end resection in single cells. An HO cut site was inserted 3.4-kb from a 10.3-kb lacO array located on Ch 2 of S. pombe. DSB formation/early resection was detected by the appearance of Rad52-mCherry foci that co-localize with the lacO array marked with LacI-GFP. Resection of the entire lacO array results in loss of the GFP signal while the mCherry signal remains. Consistent with a previous study using population based qPCR to measure resection (Langerak et al., 2011), the authors report that most resection is due to Exo1 activity and not Rqh1. Two years ago, several groups reported that Rev7 acts with 53BP1 to inhibit resection in mammalian cells. Here, the authors show that Rev7 and Crb2/53BP1 have a conserved role in preventing long-range resection in S. pombe. Furthermore, they show that Rev7 and Crb2 specifically block resection by Rqh1.Overall, the authors demonstrate that live cell imaging can be used to study end resection in single cells, and their studies show considerable cell-to-cell variation in the initiation of resection, a feature missed by population-based DNA analysis. However, I'm not convinced that the microscopy-based single-cell assay is the best way to monitor resection. The generated data imply that resection speed can be measured accurately but that relies on some assumptions.

First, we appreciate the reviewer’s determination that this single-cell LacO resection assay can be used to measure resection, including determination of cell-to-cell variability. We would like to emphasize that we expect that this assay is complementary to existing approaches (and is in no way intended to replace such methods). Moreover, we show that this assay has unique strengths (for example, information that would be missed by population studies) and future applications (e.g. the ability to look at compartmentalization and dynamics of DSBs simultaneously with measuring resection rate). Admittedly, as with any technique, there are aspects that may be better addressed with orthogonal methods. We appreciate the concerns that the reviewer has about the various parameters necessary for the assay to work robustly, and we have worked to challenge the “assumptions” with additional data, specifically with extensive qPCRbased analysis (Figures 1D, Figure 1—figure supplement 1B,C, Figure 2C, Figure 4C).

In subsection “Image analysis” the authors write "The time between the first frame with an on-target Rad52-mCherry focus and the first frame with complete disappearance of the LacO/LacI-GFP focus is the duration of resection through 3.57 kb (between the HO cut site and the start of the LacO repeats) plus the full 10.3 kb LacO array". Is resection through the complete LacO array necessary for disappearance of the GFP focus? I could imagine that the GFP focus disappears before the complete LacO array is degraded. What is the minimum number of detectable LacO repeats? The authors could integrate LacO arrays of different lengths and check what the detection limit is. Or they should at least confirm the timing of the GFP focus disappearance with the qPCR assay (According to Figure 1 Supp. 1 they could use HincII for this).

Indeed, we had carried out such analysis to determine our sensitivity to both Rad52 loading (t=0, determined experimentally, now indicated in each Figure) and “full” resection of the LacO array. This was carried out as suggested by the Reviewer through analysis of LacO arrays of decreasing lengths (Figure 1—figure supplement 3E). Based on these analyses, we expect our determination of the time at which resection initiates to be slightly AFTER the true time of resection initiation (by ~300 bps) and we expect our determination of the time at which resection through the LacO array is complete to be slightly BEFORE the true time of complete LacO resection (by <500 bps). This is based on the following observations: (1) New qPCR analysis of cells lacking Exo1 that demonstrates that the extent of resection required to visualize Rad52-mCherry at the induced DSB is ~300-500 bp (Figure 1—figure supplement 3C and Figure 1E). (2) Our ability to robustly detect a 1kb LacO array signal above background under our imaging conditions (Figure 1—figure supplement 3C) – from this further analysis, we conservatively estimate that the LacO/GFP-lacI array can be visualized until <500 bps of the array is retained. In the context of 13.9 kb of total resection distance (from HOcs to the end of the LacO array), we argue that <800 bps will only slightly influence the determined rate (<5% difference). However, as reviewer 2 correctly notes, the fact that we are detecting the initiation of resection slightly later than the true initiation time, and we are detecting the completion of resection slightly earlier than the true completion time, means that we are slightly underestimating the total duration of resection and thus slightly overestimating the rate of resection. This actually could partially explain the reason our measured rate for WT cells (median = 7.6 kb/hr) is slightly higher than previous published qPCR-based approaches, although we would predict an over-estimate maximally of 0.4 kb/hr. Indeed, the extent to which this could influence our interpretation of the data (0.4 kb/hr) is minor compared to the variability within the population, or any of the median rates that we find to be statistically significant between genotypes. Given this, rather than making assumptions, we felt that it was more transparent to use the complete distance from HOcs to the end of the LacO array in our calculations, as this is the only distance that it is possible to know with complete certainty. Lastly, as suggested by the reviewer, we have also now acquired significant qPCR-based data that supports the conclusions from this live cell assay (see below).

Also, to my eye the Rad52 focus appearance and LacI-GFP focus disappearance are not easily identified (based on the microscopy images shown). The authors should mark the frames they define as "Rad52 focus appearance" and "LacI-GFP focus disappearance".

As requested, we now indicate both Rad52-mCherry loading and GFP-lacI focus disappearance for all kymographs, as well as the resection duration and rate. Further, we now present many more examples of the population of cells, each indicating the frame that we score for “Rad52 focus appearance” in Figure 1—figure supplement 2.

In the Materials and methods section, they don't say how they define these events. Do they do it manually or using some image analysis software? What is the threshold? More details are necessary here to judge the accuracy of the method.

We now include additional details on our image analysis pipeline, which uses a combination of manual and automated routines. We would emphasize that, although aspects are carried out manually, the hundreds of individual 5D image fields from all genotypes were all pooled together, randomly sorted, and then presented to the person scoring DSB events with all identifying file name information removed. This method of blinding across ~900 image fields distributes any human error evenly across the 11 genotypes analyzed.

Also, the restriction of the assay to 5 hours by photobleaching seems to be a considerable drawback, which limits the number of "usable" cell trajectories (as shown in Figure 2—figure supplement 2B). Perhaps by changing the imaging period and number of z-stacks they could extend the time window for monitoring resection. Other approaches, such as using more photostable fluorescent proteins (e.g. LacI-mKate2, Rad52-monomeric NeonGreen), LEDs as light sources, or switching to an enzyme that cuts more efficiently than HO (I-PpoI has been successfully used in S. pombe and the efficiency of cutting is better than HO) would involve considerable more effort and time.

We invested substantial effort and time into exploring numerous induction systems, site-specific cut sites/endonuclease pairs and imaging conditions that could produce results superior to those described in this manuscript, as suggested by the reviewer. Based on this exhaustive exploration, the system described here presented the most robust and reproducible performance. Although it is likely that further improvements can still be made to this system, we would argue that our ability to characterize a wide array of genetic backgrounds to reveal new insights into regulation of DSB end resection in fission yeast highlights that the assay is of sufficient throughput. Unfortunately, it is not possible to reduce the number of z-slices without losing the ability to accurately detect sub-diffraction foci in Z or avoid loss of foci above/below the z-stack.

The authors stress that they use a single cell assay. But it is not clear what the advantage of single cell data is in their work. In the end they are comparing population medians, which could also be generated with bulk experiments.

To the contrary, we would strongly argue that the data presented here illustrate precisely why such a single cell assay provides a valuable complement to population assays. Perhaps it was not clear that the long-range resection rates are only determined from individual cells that have successfully initiated resection. For the *rev7*Δ cells, this is absolutely critical. As we show, fewer cells successfully initiate resection the absence of Rev7 (Figure 4A-C). However, once initiation occurs, long-range resection occurs much more rapidly than in WT cells (Figure 2). We would predict that these two opposite effects would be very challenging to convincingly (and certainly quantitatively) assess using a population-based resection assay (and could instead be interpreted as only one of the two behaviors, depending on which dominates in the specific assay). Indeed, our qPCR analysis, presented in Figure 4C, makes this point, with distinct effects 300 bp and 3 kb from the HO cut site. However, given *only* this information: a quite subtle early resection delay in the population (300 bp from the HO cut site) and a gain of resection 3 kb from the HO cut site in the population, it would be challenging to fully interpret this data based on qPCR alone. Thus, we think this example (unlikely to be the only case) illustrates how it will be revealing to compare results using these two approaches in future studies.

Further insights can be concluded for other genetic backgrounds, for example we can discern that loss of Crb2 does not affect resection initiation in otherwise WT backgrounds, while it increases long-range resection rates (Figure 1C and Figure 4A,B). More generally, as no DSB induction system is entirely efficient, even in budding yeast, this approach allows us to decouple early events in DSB processing from long-range resection without weighing heavily on estimates of DSB induction. Lastly, we would argue that going forward it is clear that having a cell biological assay will open up the door for analysis not possible with qPCR approaches such as DSB mobility and subnuclear compartmentalization.

The low DSB formation efficiency might be a motivation to use a single cell assay to restrict the analysis to the few cells with a DSB. However, bulk experiments generally take the cutting efficiency into account, e.g. the qPCR-based assay described by Zierhut and Diffley, (2008) considers the HO cut fraction.

While this is true, because this is population approach it does not allow individual cells to be monitored for the rate of long-range resection independently from delays in resection initiation. In the assay described here, even if there is a strong reduction in resection initiation (say 25% of WT), we can still monitor the rate of long-range resection in that 25% of cells; again, this cannot be assessed by qPCR. Indeed, cells lacking Rev7 are a good example of how these two phases can be analyzed distinctly.

Rad52 foci are used as a read out for DSB formation and initiation of resection. Can these two steps be separated, for example, by measuring DSB formation by qPCR using primers flanking the HO cut site on the same samples used for the ApoI protection assay, or possibly using Mre11-mCherry instead of Rad52? It appears from Figure 1B (upper panel) that resection through most of the lacO array is required for a strong Rad52 signal.

In the revised manuscript, we include our further characterization for the extent of resection required to load sufficient Rad52-mCherry to be robustly visualized in the assay. This was achieved, as suggested by the Reviewer, through comparison of live cell imaging and ApoI protection/qPCR, which can be found in Figure 1—figure supplement 3. Based on this, we now estimate that ~300 bps of resection are required.

In theory, the HO-induced DSB is unrepairable, but because the HO cutting efficiency is quite low, and S. pombe cells are in G2 most of the time, if one sister chromatid was cut and engaged in repair with the uncut sister it could result in an underestimation of resection. Were cells with a transient Rad52 focus detected? The exo1 example in Figure 1 appears to have a Rad52 focus that appears early and then goes away. Does elimination of Rad51 change the number of cells resecting or rate of resection? At late times, when the mCherry signal is very bright, are both sister chromatids cut and resected?

We agree that this is an important point and feel that there are several observations that can speak to this concern. We have not observed examples of WT cells (or even mutant cells) in which Rad52 associates with the LacO array and is then lost, followed by repair; in this case we would expect the cells to avoid checkpoint arrest, and proceed into mitosis. In the example the Reviewer points out, cells lacking Exo1 (Figure 1E), any cell that recruits Rad52 fails to resect, but remains checkpoint arrested, suggesting failed repair rather than repair of a single DSB using the sister as a template. This interpretation is further supported by the observation that loss of Crb2 in this background fully recovers Rad52 loading (Figure 4A,B). To further address the frequency at which both sisters are cut, we analyzed Rad52 foci upon cell division in *crb2*Δ cells, which proceed into mitosis due to the checkpoint defect associated with this allele. We found that in the vast majority of cases (21/27), both daughter cells inherited a fully resected site-specific DSB (see Author response image 2). In the remaining six cells, it is challenging to interpret because they proceeded into mitosis prior to full resection. From this, we also conclude that the inefficiency of DSB induction relates either to insufficient HO expression prior to Sphase or permissive nucleosome positioning, as all of our data are reported as the frequency of DSB induction per cell cycle observed.

**Author response image 2. respfig2:** Lineage tracing of *crb2*Ä cells (which are checkpoint-deficient) reveals that the vast majority of sisters arising by division (visualized as splitting of one circle to two) both inherit a fully resected LacO array (pink).

Langerak et al., used a qPCR assay to measure end resection and reported no defect in resection initiation (35 nt from HO cut site) in the exo1 mutant or exo1 rqh1 double mutant. Is the failure to detect Rad52 foci in most exo1 cells because resection tracts are <90 nt or because the amount of ssDNA required to support a Rad52 focus is much longer than 90 nt? Given that the Rad52 single is quite weak until the lacO array disappears, I think the authors might be under-estimating the amount of ssDNA to visualize a Rad52 focus. Does the exo1 mutant show normal DSB formation (measured with primers flanking the HOcs) and resection to the ApoI site located 168 nt from HOcs? Similarly, is DSB formation normal in the rev7 mutant and can early resection be detected by the qPCR assay?

As stated in the manuscript, we experimentally estimate that at least 30 copies of Rad52 must be loaded onto the resected DNA to be visualized. Unfortunately, the number of Rad52 copies that are expected to remain associated with the remodeled nucleoprotein filament has not been clearly established experimentally. We initially set lower bounds as 90 nts. To provide further insight, we carried out qPCR experiments on cells lacking Exo1 and Ctp1 using the approach described in Langerak et al., as suggested by the Reviewer, but with many more primer sets. All genetic backgrounds (including cells lacking Exo1 or Ctp1) have similar levels of resection 168 bps from the HO cut site (not shown). Most useful is the finding that we do see a substantial resection defect in cells lacking Exo1 using primers 300 bps from the HO cut site, almost to the extent seen in cells lacking Ctp1 (Figure 1—figure supplement 3C). Moreover, we see no resection 3 kb from the DSB at 90’ post-induction. Taken together we estimate that cells lacking Exo1 have a block between ~150 bps and ~300 bps of resection based on these population data. Importantly, we do visualize transient Rad52 loading in cells lacking Exo1 (Figure 1E), consistent with an upper bounds of ~300 nts of resected DNA being sufficient to load enough Rad52-mCherry copies to be visualized. However, we have given a broader range for the extent of resection necessary to observe robust Rad52-mCherry recruitment as between 150-300 bps in the revised manuscript. The effect of loss of Rev7 by qPCR is now included in Figure 4C. Again, consistent with our initial interpretation, there is a slight defect in resection 300 bps from the HO cut site, but a gain in resection 3 kb away. This orthogonal approach therefore supports our hypothesis that Rev7 promotes early steps in resection but has a gain in long-range resection rate (once this early phase is overcome).

Why does rqh1 suppress the early resection defect of rev7? An odd result that is not discussed in the text.

We do not yet have a molecular understanding of how loss of Rqh1 is able to suppress the early loss of Rad52-mCherry loading in cells lacking Rev7. We do acknowledge this in the revised text, but further insight will require additional experimentation, which is ongoing.

While the overall findings on the role of Rev7 and Crb2 repressing the Rqh1 resection pathway are certainly of interest to the field, the data analysis needs to be improved.

We appreciate that the reviewer found the findings revealed by this work to be of interest and have endeavored to address the remaining concerns regarding image analysis and validation using orthogonal assays in this revision.

Reviewer #3:DNA end resection is a process that initiates recombination-based DNA double strand break repair. As resection also generally inhibits non-homologous end-joining, regulation of DNA end resection is an important process. The human 53BP1 protein, though its various effectors, has been found to be an inhibitor of DNA end resection, although mechanistic insights are lacking. These processes appear to be at least partially conserved in low eukaryotes.The authors are using a S. pombe as a model system, where they developed an assay allowing the monitoring of resection in live cells (Figure 1). The assay is based on the disappearance of GFP-LacI and appearance of Rad52-mCherry signal next to HO-endonuclease induced DSB. This is an interesting method that will be useful for high throughput microscopy-based screenings. However, the method has certain limitations in contrast to established southern blotting, PFGE or RTPCR-based methods:

We appreciate the reviewer’s assessment of the value of a microscopy-based resection assay. While we agree that the assay has some weaknesses compared to previously developed population-based assays, we would argue that qPCR and Southern blot approaches also have limitations in contrast to the microscopy-based method developed here (see also response to reviewer 2). In the larger scope, we expect that both assays will provide complementary insights into the control of DSB end resection.

a) The assay measures Rad52 accumulation, which is a step after resection. Rad52 is a mediator that loads Rhp51 on RPA-coated resected DNA. Using RT-PCR based assay, the authors established that there is a correlation between both processes (resection and RAD52 loading). However, this was only done in wt background, and it is possible RAD52 loading might differ in the mutants analyzed, and Rad52 might be loaded with different kinetics dependent on the resection pathway. The authors calculate resection rates making the assumption that there is no difference.

In the revised manuscript, we include substantial additional support for our conclusion by analyzing additional genetic backgrounds (*exo1*Δ, *rev7*Δ, *ctp1*Δ) using the qPCR based method (Figure 1D, Figure 2C, Figure 1—figure supplement 3C, Figure 4C). Our findings support the notion that the extent of Rad52-mCherry loading is coincident with resection of at least 150-300 bps of DNA flanking the DSB (Figure 1—figure supplement 3C and also see response to reviewer 2).

b) The assay measures resection of DNA bound by GFP-LacI, a non-physiologic binder. Therefore, the resection proteins must displace LacI for resection to occur. Under physiologic conditions, DNA near DSBs will likely be chromatinized and subsequently remodeled allowing resection. This is especially a concern when analyzing the role of chromatin binders such as 53BP1/Crb2. Also, the individual resection pathways may be affect by LacI binder to a different degree, which complicates interpretation.

We agree that in principle the LacI binding could alter how resection proceeds through the array. In addition to the experimental evidence presented in the initial manuscript supporting the validity of resection through the array, we now also include qPCR analysis on the other (non-LacO containing) side of the HO nuclease cut site; this analysis supports the conclusion that the LacO array does not strongly influence resection (Figure 1D). This concern has also been explored by more broadly including analysis of resection using the orthogonal, qPCR based assay (Figure 1D, Figure 2C, Figure 1supplemental figure 3C, Figure 4C).

The most interesting finding is that Rev7 and 53BP1/Crb2 appear to repress long-range resection dependent on Rqh1 (Figure 2): In the absence of Rev7 and Crb2, long-range resection is accelerated, which is independent of Exo1 and depends on Rqh1.1) This observation should be verified using a previously established assay, given the concerns listed above. Related to this point, it remains a formal possibility that the action of Rqh1 is not specific to resection, but to strip LacI, allowing resection through another process. Performing resection with the same mutants in an established setup (with no LacI) will address this concern as well.

In the revised manuscript, in addition to the validation by the qPCR assay that the LacO array does not alter resection (Figure 1D), we provide further evidence that cells lacking Rev7 show more rapid long-range resection using the qPCR approach (Figure 2C, Figure 4C). These data strongly support that resection, rather than some other activity (such as loss of LacI) is responsible for the progressive loss of GFP-LacI signal.

The authors then go on to demonstrate differential effects of Rev7 and Crb2 on early resection steps, based on timing of Rad52 foci appearance upon break induction (Figure 3). I am concerned about these results: using Rad52 (a protein with a function downstream of resection) does not appear to be correct as a marker of "early" resection.2) The results should be analyzed in an established assay where the readout is clearly early resection. Also, to make claims on early resection, the analysis should include mutants deficient in short-range resection (e.g. mre11) and mutants in long-range resection, where both pathways have been inactivated (e.g. exo1 rqh1).

The revised manuscript includes substantial new data to address how to place Rad52 loading in the context of resection progression. We now leverage cells lacking Exo1 to obtain a more precise estimate of the extent of resection required to load Rad52. By comparing qPCR analysis (Figure 1—figure supplement 3C) and our image analysis (Figure 1E, Figure 4A,B), we now estimate that ~200-300 bp of resection is necessary to visualize Rad52-mCherry loading at the DSB. Moreover, we compare cells lacking Exo1 with those lacking CtIP/Ctp1 (a “true” resection initiation factor); these two genetic backgrounds have very similar resection defects 300 bp from the HO cut site. We would therefore argue that while Exo1 is not required for “resection initiation”, it is required for the early phase of long-range resection (defined here as the earliest phase in which Exo1 is required).

3) In Figure 2, the authors demonstrate that rev7 mutants have accelerated long-range resection, which is dramatically decreased when rqh1 is additionally mutated (rev7 rqh1), supporting the hypothesis that resection in rev7 mutants is Rqh1 dependent. In contrast however, the in Figure 3, the "early" resection is increased when rqh1 is mutated in rev7 background, a completely opposite effect. This is very confusing. The authors comment that "early" resection is a combination of Mre11 and Exo1/Rqh1 dependent processes. This dichotomy reinforces my concerns about the robustness of the experimental setup.

As stated in our response to reviewer 2, it is true that at present we do not fully know how to interpret this result, which will require further experimentation (likely using a number of additional approaches) to dissect. However, with regards to the general robustness of the assay, we believe that the revised manuscript provides substantial new support for the experimental setup and the validity of the interpretations made from the data.

In summary, the manuscript presents an interesting assay and interesting pieces of data, but it seems rather preliminary at this point.

We have endeavored to convince the reviewer that, with the addition of supportive data from orthogonal assays and further validation, the manuscript is now sufficiently developed to support publication.

[Editors' note: the author responses to the re-review follow.]

The biological findings of this manuscript are interesting, particularly the role of Rev7 in suppressing resection by the Rqh1 pathway, but there are significant concerns over the reliability of the assays used. The authors place equal importance on the fluorescence-based resection presented; however, the reliability of the assay is questionable and the weak point of this work. The conceptual idea of the assay is good, but there are reservations about its implementation and use to quantify resection speed. The authors have, for the most part, established the utility of their assay for the analysis of long-range resection, although they still need to provide a graphical example of their quantification.

We are gratified that the biological findings of the manuscript were found to be of interest and that there was enthusiasm for the utility of our assay. The remaining point, the need for graphical examples of the quantification, has been addressed in the revised manuscript – please see Figure 1D,E, Figure 1—figure supplement 2, Figure 2B, and Figure 2—figure supplement 1.

They discovered interesting roles of Rev7 and Crb2 in Rqh1-dependent resection. As both a methods contribution and a contribution to the recombination field, these findings could justify publication of a revised manuscript.

We appreciate that the insights into the mechanisms by which Rev7 and Crb2 influence long-range resection (in an Rqh1-dependent and Exo-independent manner) were found to be of interest, therefore warranting further consideration of our manuscript.

However, the manuscript then goes on to dissect the roles of these proteins in proximal recombination. Here the work is both incomplete and internally inconsistent. It would appear that the data in Figure 4A and Figure 4C are inconsistent with one another: 4A show a reduction in "Cell cycles with Rad52 foci" for rev7delta, whereas 4C shows no change, within error, for rev7delta at 300 bps [The figure legend, as well as the associated text in the main body of the manuscript, is not justified: "The extent of resection 300 bps and 3 kb from the HO cut site as assessed by the restriction enzyme/qPCR method supports less efficient resection initiation in rev7Δ cells compared to WT at 300 bp…"].

We acknowledge the concerns of the reviewers with respect to the measurement and interpretation of data corresponding to proximal resection. As we have not obtained additional data to clarify these points, we have removed this section of the manuscript (previously Figure 4 and supplements). This revision is therefore focused solely on the long-range resection assay and the novel finding that Rev7 acts as an inhibitor of RecQ helicase-mediated resection.

Also, the conclusion that of Rev7 promotes early resection is based entirely on how quickly a Rad52 focus forms after break induction, and how many cells have visible Rad52 foci. It was suggested that the authors use a more direct measurement of break induction by qPCR using primers flanking the HO cut site. This has not been done and is a significant concern.

Although we have removed the data and text related to proximal resection, the revised manuscript includes qPCR analysis of the efficiency at which the site-specific DSB is induced in the population. The addition of this data supports the assertion that resection of the site-specific DSB takes place in the vast majority of fission yeast in S/G2 with a DSB, as the fraction of cells that (1) show loss of the qPCR product across the HO cut site (Figure 1—figure supplement 3a, at 180’ after addition of uracil), (2) undergo resection 168 bps from the HO cut site (Figure 1—figure supplement 3C, at 180’ after the addition of uracil), and (3) ultimately load Rad52-mCherry (Figure 1—figure supplement 3C, at 180’ after addition of uracil) are similar (~15-20%).

If one compares the qPCR data for resection 168-bp from the HO cut site shown in Figure 1D and Figure 1—figure supplement 3B there are vast differences – 3% resection at 120 min in one figure compared with 15% resection at 120 min in the other. If there is this much variability between populations of cells then it could explain why the rev7 mutant looks different to wild type, and why rqh1 appears to suppress the early resection defect of rev7. How many independent inductions were performed for image analysis?

In this revision, we include population data (qPCR) to test the reproducibility of DSB induction; these data (Figure 1—figure supplement 3A and Figure 2E) demonstrate that HO nuclease has ~20% efficiency from multiple biological replicates. As detailed in Supplementary file 2 for all genotypes, the data for the image analysis is from 6 independent inductions (WT) and 5 independent inductions (rev7Δ). More generally, there is substantial evidence that the differences between resection rates that we measure in different genotypes are highly reproducible.

The authors need a reliable method to assess DSB formation independent of Rad52 focus formation before drawing conclusions about a role for Rev7 in promoting early resection.

Again, although we have removed the data/interpretation related to a role for Rev7 in proximal resection, we do now include qPCR analysis across the HO cut site that supports the conclusion that Rev7 does not influence the efficiency of DSB induction (Figure 2E).

Furthermore, In Figure 1D, the authors use a qPCR assay to detect formation of ssDNA 168 and 14,253 bp from the HO cut site. From this assay, very few (2%) cells exhibit resection to the end of the lacO array 360 min after HO induction. This would appear to contradict the microscopy assay, which shows resection through the array takes ~150 minutes.

We understand that this appears contradictory at first glance, but importantly the time courses for the qPCR (population experiment) and the measurement of the time to resect the array (in single cells) cannot be directly compared. The “time 0” for the qPCR experiments is the addition of uracil to the media. Thus, there is a substantial lag (between 60 and 240 min, which can be seen in the red curve in Figure 1supp3C) for cells to express the HO nuclease, reach S phase (the only time in the cell cycle when we observe that HO can act on its target site, Figure 1B), and initiate resection. For the single cell assay we isolate measuring the rate of long-range resection by setting the “time 0” to the point at which Rad52-mCherry has visibly loaded. Thus, on the surface this apparent “contradiction” instead reflects that the qPCR data is influenced by many rates of sequential steps (HO expression, cell cycle/access of HO, resection initiation). Indeed, this comparison highlights yet another the advantage of the assay described in the manuscript, which specifically interrogates long-range resection without these other confounding steps contributing to the measured rate. We have also edited the text to clarify this point.

Also, data should really be from biological replicas, not technical replicates of qPCR.

We agree, and these data include multiple biological replicates (as is now clear from the figure legend for the measurements of HO cut site induction in Figure 2E).

Consequently, whether or not Rev7 has a role in proximal resection is not clear from the authors' data.

Again, we have removed this aspect of the manuscript.

There was agreement that manuscript needs to be revised:1) It is essential that the confusing data on short range resection are clarified or removed from the manuscript. If the proximal resection data are retained, then the comments raised above need to be addressed. In addition, in the review of the prior version of this manuscript, reviewer #3 commented on this part in the original submission: "Also, to make claims on early resection, the analysis should include mutants deficient in short-range resection (e.g. mre11) and mutants in long-range resection, where both pathways have been inactivated (e.g. exo1 rqh1)." The authors did not address this previous request for clarification of this unexpected finding in this revised manuscript, using the analyses described for proximal resection (i.e., Figure 4).It is unclear whether the authors can make these revisions within the timeframe given. If not, then these data on proximal resection would need to be removed, and the conclusions of the manuscript refocused. One major finding is the effect of Crb2 on long-range resection: while this is new in S. pombe, it is well-described in S. cerevisiae (Rad9). The second major finding is the effect of Rev7 on long-range resection: this was only shown in humans, but not in the microbial eukaryotes. Although significant, the impact of the current manuscript is diminished without the proximal resection data; consequently, the author would need to make the contributions of their sound work much clearer in an expanded Discussion section of their work in relation to the existing literature (in any event, the existing Discussion section is inadequate.).

We agree with points of the reviewers with respect to the proximal resection data and have removed this from the manuscript. Further, we acknowledge (and now discuss in greater detail in the discussion) that an influence of Rad9 on long-range resection in budding yeast has been described previously. Further, data argue that this effect could involve Sgs1, the Rqh1 orthologue. However, we would argue that the following conclusions coming from this study are unique and impactful:

1) An entirely novel (and indeed the key) aspect of our study is that Rev7 acts to inhibit long-range resection through the RecQ helicase (Rqh1) pathway. Although it is true that Rev7 was suggested to inhibit resection in mammalian cells, these measurements were largely indirect, and the phase of resection influenced by Rev7 was not addressed.

2) We show that the influence of Crb2 on long-range resection is also through repression of Rqh1. This is consistent with budding yeast studies but is clearly shown to occur through a specific effect on long-range resection in our study.

We have articulated these points more clearly in the revised Discussion section.

2) The authors use appearance of a Rad52-mCherry focus to identify onset of resection and disappearance of a lacO/LacI-GFP focus to identify resection past a site some 14 kb away from the DSB site. The problem is that both signals are rather fuzzy in many of the image series shown. This is especially true for the Rad52-mCherry focus (see e.g. the lower image series in Figure 1B, the first three image series in Figure 1—figure supplement 2, Figure 2A). The decision if a signal is judged as a focus or not is crucial, as it is the basis to calculate resection speed. This decision is made manually. Although the authors try to "equalize" the error by randomizing the image series prior to analysis, it seems to be a quite ambiguous approach of questionable reliability. What is their criterion to judge if a focus is present or not? Do they compare with background control strains? Why don't the authors use software to quantify the fluorescence signals and generate intensity trajectories? There are several non-commercial image analysis software packages available dedicated to exactly this purpose. Based on thresholds defined by appropriate control strains, appearance and disappearance of the signals could then be identified in a more controlled and rational way. A minimal requirement is that the author provides an x-y graph of foci fluorescent intensity vs time for each their examples of time-lapse video data in the manuscript.

We evaluated the utility of several image analysis software (including the TrackMate ImageJ plugin, CellProfiler, u-track 2.0, and even custom Matlab code). Indeed, we are experts at developing quantitative image analysis routines (please see our previous, peer-reviewed works – Schreiner et al., 2015 and Zhao et al. 2016), but all were inadequate due to the uniquely stringent criteria that we placed on identifying a Rad52-mCherry focus (described in the Methods), which goes well beyond simply an intensity measurement. Nonetheless, to satisfy the reviewer’s request, we include data to address the issue of “background control strains” now in Figure 1—figure supplement 3D-G. As you can see, there is little to no chance that background fluorescence or “noise” could reproducibly contribute to a focus assignment. Moreover, as stated in the manuscript, the frequency of identifying a Rad52-mCherry focus that meets our criteria (co-localized with the array for two time points) and found that this can essentially never be observed if HO nuclease is not expressed (“Importantly, we do not observe loading of Rad52-mCherry at the LacO/LacI-GFP array in the absence of HO endonuclease expression (on-target Rad52 foci in < 0.2% of uninduced cells, n=657)”). We do recognize that it is imperative to that we can robustly identify these events, and ideally to provide a quantitative framework for how we do so. To that end, as requested, we include fluorescent intensity versus time plots in the revised manuscript. Depicting the data in this manner makes it qualitatively quite clear, for example, that cells lacking Rev7 resect the array much more quickly than WT strains.